# Reversible fold-switching controls the functional cycle of the antitermination factor RfaH

Philipp Konrad Zuber[1], Kristian Schweimer[1], Paul Rösch[1,2], Irina Artsimovitch[3,4] & Stefan H. Knauer [1]

RfaH, member of the NusG/Spt5 family, activates virulence genes in Gram-negative pathogens. RfaH exists in two states, with its C-terminal domain (CTD) folded either as α-helical hairpin or β-barrel. In free RfaH, the α-helical CTD interacts with, and masks the RNA polymerase binding site on, the N-terminal domain, autoinhibiting RfaH and restricting its recruitment to *ops*DNA sequences. Upon activation, the domains separate and the CTD refolds into the β-barrel, which recruits a ribosome, activating translation. Using NMR spectroscopy, we show that only a complete *ops*-paused transcription elongation complex activates RfaH, probably via a transient encounter complex, allowing the refolded CTD to bind ribosomal protein S10. We also demonstrate that upon release from the elongation complex, the CTD transforms back into the autoinhibitory α-state, resetting the cycle. Transformation-coupled autoinhibition allows RfaH to achieve high specificity and potent activation of gene expression.

[1] Lehrstuhl Biopolymere, Universität Bayreuth, Universitätsstraße 30, 95447 Bayreuth, Germany. [2] Forschungszentrum für Bio-Makromoleküle, Universität Bayreuth, Universitätsstraße 30, 95447 Bayreuth, Germany. [3] Department of Microbiology, The Ohio State University, Columbus, OH 43210, USA. [4] The Center for RNA Biology, The Ohio State University, Columbus, OH 43210, USA. Correspondence and requests for materials should be addressed to I.A. (email: artsimovitch.1@osu.edu) or to S.H.K. (email: stefan.knauer@uni-bayreuth.de)

Multi-subunit RNA polymerases (RNAP) transcribe all cellular genomes and interact with a plethora of accessory proteins that modulate every step of RNA synthesis. Among them, NusG/Spt5 is the only regulator that is conserved across all domains of life[1]. NusG homologs control gene expression by reducing RNAP pausing and arrest to enhance its processivity[2–4] and by enabling crosstalk between transcription and concomitant cellular processes. These proteins physically link elongating RNAP to a ribosome[5] or transcription termination factor Rho[6,7] in bacteria and to factors mediating mRNA capping[8], histone modification[9], and somatic hypermutation[10] in eukaryotes.

The modular structure of NusG proteins underpins this bridging activity (Fig. 1a). The N-terminal domains (NTDs) exhibit mixed α/β topology and establish similar contacts to the two largest subunits of bacterial, archaeal, and eukaryotic RNAPs[11–15]. In bacteria, these contacts are mediated by the β′ clamp and the β lobe and protrusion domains[13]. The C-terminal domains (CTDs; one in bacteria and archaea, multiple in eukaryotes) contain a Kyrpides, Ouzounis, Woese motif[16], fold into a five-stranded β-barrel that is flexibly connected to the NTD, and serve as interaction platform for various binding partners, making co-transcriptional contacts to cellular proteins that ultimately determine their effects on gene expression. In *Escherichia coli*, NusG-CTD interacts with Rho to inhibit synthesis of foreign and aberrant RNAs[6,17] or with ribosome to couple transcription to translation[5,18].

In addition to housekeeping factors that co-localize with elongating RNAP across most genes[19–21], highly specialized NusG paralogs are present in ciliates[22], plants[23], and bacteria[24]. In order to avoid off-target recruitment, these paralogs must be specifically recruited to their target genes. This is particularly critical when their function is opposite to that of housekeeping NusG, as is the case of bacterial paralogs which silence Rho-dependent termination[25].

RfaH, the best studied NusG paralog, activates expression of cell wall biosynthesis, conjugation, and virulence genes by inhibiting Rho[26]; mutations in *rho* and *nusG* suppress the loss of *E. coli rfaH*[27]. RfaH activates fewer than ten operons in *E. coli*, each containing an operon polarity suppressor (*ops*) element in their 5′ UTRs[28]. When RNAP pauses at the *ops* site, the non-template (NT) DNA strand in the transcription bubble forms a hairpin structure[13,29]. During recruitment, RfaH makes base-specific contacts with two flipped-out *ops* bases via its NTD. While these contacts explain sequence specificity of RfaH, off-target recruitment and competition with housekeeping NusG is additionally controlled by autoinhibition.

RfaH-NTD exhibits the mixed α/β topology typical for NusG proteins but, in contrast to all other known NusGs, the RfaH-CTD folds as an α-helical hairpin in free RfaH (all-α state; Fig. 1b). The CTD hairpin tightly interacts with the NTD, masking the RNAP-binding site and autoinhibiting RfaH[30]. The relief of autoinhibition requires domain dissociation, thought to be triggered by transient contacts to *ops*. Thereafter, the released NTD binds to the clamp helices of the β′ subunit (β′CH) and the gate loop of the β subunit (βGL) of the RNAP[26,28], while the CTD spontaneously and completely refolds into a NusG-like β-barrel (all-β state; Fig. 1b) and recruits the ribosome via interactions with ribosomal protein S10, substituting for a missing Shine-Dalgarno sequence[31]. As striking as this transformation is, the lack of spurious RfaH recruitment at non-*ops* sites[28] suggests that refolding may be reversible: following dissociation from RNAP at a terminator, RfaH must either perish or transform back into the autoinhibited state[32] because activated RfaH does not require *ops* for recruitment[30,33].

Here, we used NMR spectroscopy adapted to supramolecular, multicomponent systems in combination with functional studies to explore the conformational transitions that accompany RfaH binding to and dissociation from RNAP. Our results indicate that RfaH functions in a true cycle. We identify the *ops*-paused transcription elongation complex (EC) as a minimal activation signal for RfaH and demonstrate that, upon recruitment, RfaH-CTD refolds into the β-barrel that subsequently interacts with ribosomal protein S10. We further show that RfaH-CTD refolds into the α-helical state after RfaH release from the EC, thereby re-establishing the autoinhibited state. Our results demonstrate unmatched conformational and functional plasticity of RfaH, which refolds not once but twice during its functional cycle, as befits a transformer protein[34].

## Results

**The *ops*-paused EC is necessary for RfaH activation.** To elucidate the mechanism of RfaH recruitment to the EC and identify a signal that induces domain opening, we used a combination of solution-state NMR spectroscopy approaches which allow the characterization of protein:ligand interactions and structural transitions as well as the analysis of excited, low-populated states. In addition to uniformly $^{15}N$-labeled proteins, samples were employed where [$^1H$,$^{13}C$]-labeled methyl groups of Ile, Leu, and Val residues in perdeuterated proteins served as NMR-active probes ([I,L,V]-labeling); the latter method increases the sensitivity to enable studies of large complexes[35].

In the methyl-transverse relaxation optimized spectroscopy (methyl-TROSY) spectrum of free [I,L,V]-RfaH we observed only signals of the autoinhibited form (Fig. 2a). To test if, nevertheless, free RfaH exists in equilibrium of the closed and open conformations, with the open state being only low-populated, we first carried out $^{15}N$-based chemical exchange saturation transfer (CEST) experiments (Supplementary Figure 1a-d; ref. [36]). This method allows for the detection and characterization of 'invisible', i.e., sparsely populated, excited conformational states that are in slow chemical exchange with a visible ground-state conformation. In CEST experiments the saturation of $^{15}N$ spins by a weak radio frequency field can be transferred between different conformational states if these states exchange on a timescale of 5–50 ms. If the transmitter frequency, which is stepped through the spectral region of $^{15}N$ spins, coincides with the resonance frequency of a spin, the signal intensity is significantly decreased, causing a dip in the CEST profile (normalized intensity ($I/I_0$) of a signal as function of the transmitter frequency of the saturation field). If the major state is in equilibrium with another (minor) state, the exchange

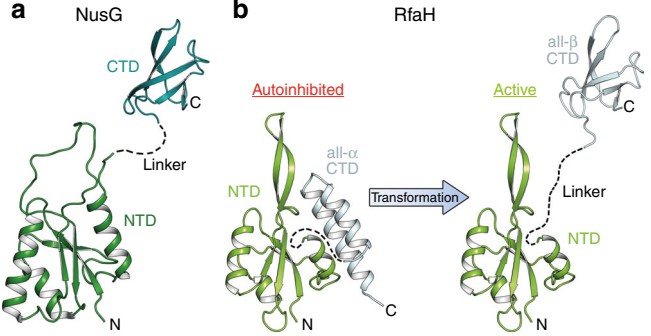

**Fig. 1** Structures of NusG and RfaH from *E. coli*. Structures of **a** NusG and **b** RfaH are in ribbon representation, the linker connecting the domains is indicated by a dashed line. PDB IDs: NusG-NTD, '2K06'; NusG-CTD, '2JVV'; RfaH, '5OND'; RfaH-CTD, '2LCL'

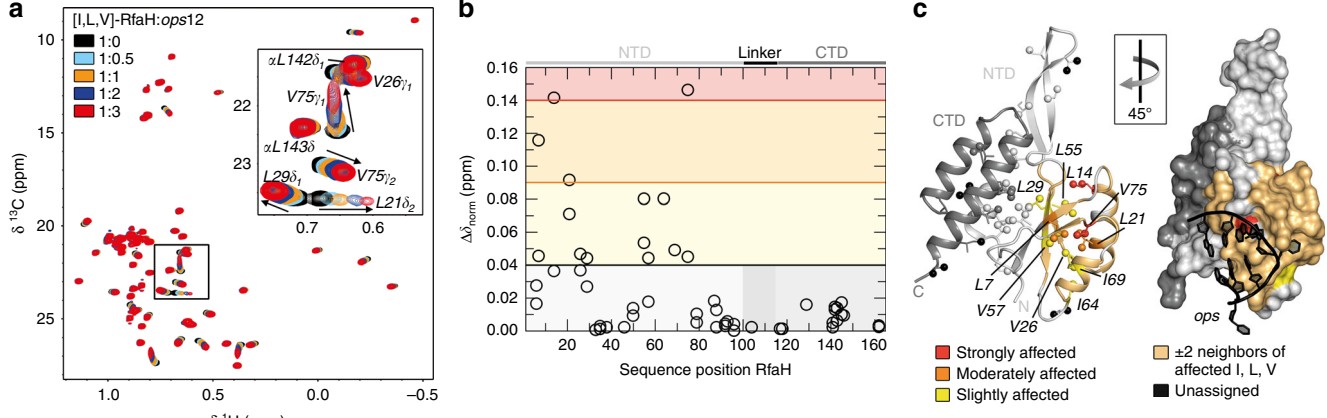

**Fig. 2** RfaH:*ops* interaction. **a** 2D [$^1$H, $^{13}$C] methyl-TROSY spectra of 45 μM [I,L,V]-RfaH titrated with *ops* (concentration of stock solution: 1.3 mM). Inset: enlargement of boxed region. **b** Interaction of [I,L,V]-RfaH with *ops*. Methyl-TROSY-derived normalized chemical shift changes vs. sequence position of RfaH (corresponding spectra are depicted in **a**). Horizontal lines: significance levels of $\Delta\delta_{norm} = 0.04$ ppm, black; = 0.09 ppm, orange, = 0.14 ppm, red. Source data are provided as a Source Data file. **c** *ops* binding surface of RfaH as derived from the titration of [I,L,V]-RfaH with *ops*. Affected methyl groups are mapped onto the RfaH:*ops*9 structure (PDB ID: '5OND'). RfaH is shown in ribbon (left) and surface (right) representation (RfaH-NTD, light gray; RfaH-CTD, dark gray), *ops*9 in ribbon representation (black) with nucleosides as sticks. The arrow indicates how the structures are rotated with respect to each other. Termini are labeled. For graphical representation of the interaction site the whole amino acid is colored. Ile, Leu, and Val residues are shown as sticks with the carbon atoms of the methyl groups as spheres. Slightly affected ($0.04 \leq \Delta\delta_{norm} < 0.09$ ppm), yellow; moderately affected ($0.09 \leq \Delta\delta_{norm} < 0.14$ ppm), orange; strongly affected ($\Delta\delta_{norm} \geq 0.14$ ppm), red; unaffected, colored according to their domain; not assigned methyl groups, black. Two amino acids on either side of an affected Ile/Leu/Val residue are highlighted in beige unless they were unaffected Ile/Leu/Val residues

between these states will be evidenced by a second dip in the CEST profile occurring at the resonance frequency of the minor state. In the CEST profiles of the RfaH-CTD residues that have well-separated signals in the all-α and the all-β state, no second dip at the expected chemical-shift position corresponding to the all-β state could be observed (Supplementary Figure 1a-d). Thus, within the detection limits of the CEST experiment (population > 0.2%, exchange rate 20–200 s$^{-1}$), all free RfaH occupies the closed, autoinhibited state.

To test whether RfaH opens and closes on a timescale faster than accessible by CEST experiments, we performed Carr-Purcell-Meiboom-Gill sequence (CPMG) experiments. This analysis enables the measurement of the contribution of chemical exchange to the transverse relaxation rate ($R_2$) of any nucleus for exchange processes in the range of ~200–2000 s$^{-1}$ [37]. In brief, a series of refocusing 180° pulses is applied with different time intervals ($\tau_{180}$) between the pulses. During long time intervals, i.e., at low CPMG frequencies ($1/2\tau_{180}$), the chemical exchange can contribute to $R_2$, resulting in an increase of $R_2$. In the CMPG experiments of $^{15}$N-RfaH, we observed slightly enhanced $R_2$ rates at lower CPMG frequencies for residues located in a loop in the DNA-binding region[29] (T72, V75) as well as in the β-hairpin (S47), suggesting that these RfaH-NTD regions exhibit flexibility (Supplementary Figure 1e). In contrast, $R_{2,eff}$ did not change for RfaH-NTD residues in the domain interface (F51) or RfaH-CTD residues (F123, G135, M140, L145; Supplementary Figure 1e), indicating that the autoinhibited state is stable. Together, these findings argue against an equilibrium of the closed and open conformations of RfaH on a timescale faster than 0.5 ms–50 ms.

The DNA-binding site of RfaH is located on the RfaH-NTD, opposite the RfaH-CTD interaction surface[13,29]. A [$^1$H,$^{15}$N]-heteronuclear single quantum coherence (HSQC)-based NMR titration of $^{15}$N-RfaH with *ops* indicated that binding of RfaH to *ops*DNA does not induce domain separation[29]. Exploiting the high sensitivity of methyl groups, we next wanted to corroborate this result. Chemical-shift changes upon titration of [I,L,V]-RfaH with *ops*DNA were consistent with the DNA-binding site determined via the $^{15}$N-based titration and observed in the

RfaH:*ops*9 crystal structure and in the cryo electron microscopy (EM) RfaH:*ops*-paused EC structure (Fig. 2; refs. [13,29]). Signals corresponding to the all-β RfaH-CTD could not be observed during the titration, suggesting that binding to DNA alone cannot be a signal for domain opening.

RfaH weakly binds to free RNAP[38]. To test whether these contacts could activate RfaH, we measured one- and two-dimensional (1D, 2D) methyl-TROSY spectra of [I,L,V]-RfaH titrated with RNAP (Fig. 3a). The overall intensity of signals corresponding to autoinhibited RfaH decreased uniformly, but no changes in chemical shifts were observed and β-barrel CTD signals did not appear (Fig. 3a and Supplementary Figure 2), indicating that while RfaH can interact with RNAP, this binding does not induce domain dissociation/transformation. Adding an excess of NusG-NTD to the [I,L,V]-RfaH:RNAP complex recovered some of the intensity of [I,L,V]-RfaH signals (Fig. 3b), implying RfaH displacement by NusG-NTD. Since RfaH and NusG share binding sites[13,28], this finding suggests that the closed RfaH binds near the final RfaH-NTD binding site on the EC.

These results show that neither DNA nor RNAP alone can relieve RfaH autoinhibition. To test if EC paused at the *ops* site (*ops*EC) is suffient to induce domain separation, we assembled *ops*EC with a nucleic-acid scaffold (Supplementary Figure 3) and perdeuterated RNAP. A methyl-TROSY-based titration of [I,L,V]-RfaH with the *ops*EC showed that signal intensity of [I,L,V]-RfaH methyl groups decreased non-uniformly, with only slight chemical shift changes (Fig. 4a). Signals of the α-helical RfaH-CTD disappeared while, concurrently, resonances corresponding to the β-barrel RfaH-CTD appeared and gradually intensified, indicating refolding of the RfaH-CTD (Fig. 4a and Supplementary Figure 4). Next we wanted to exclude the possibility that the RfaH-CTD refolding is due to cleavage of the RfaH linker caused by protease impurities or sample degradation during long-lasting NMR experiments. Both scenarios would lead to the release of the RfaH-CTD and its subsequent spontaneous transformation, as shown for an RfaH variant where a TEV protease cleavage site was introduced into the linker[31]. Thus, we performed translational diffusion experiments of [I,L,V]-RfaH and [I,L,V]-RfaH-

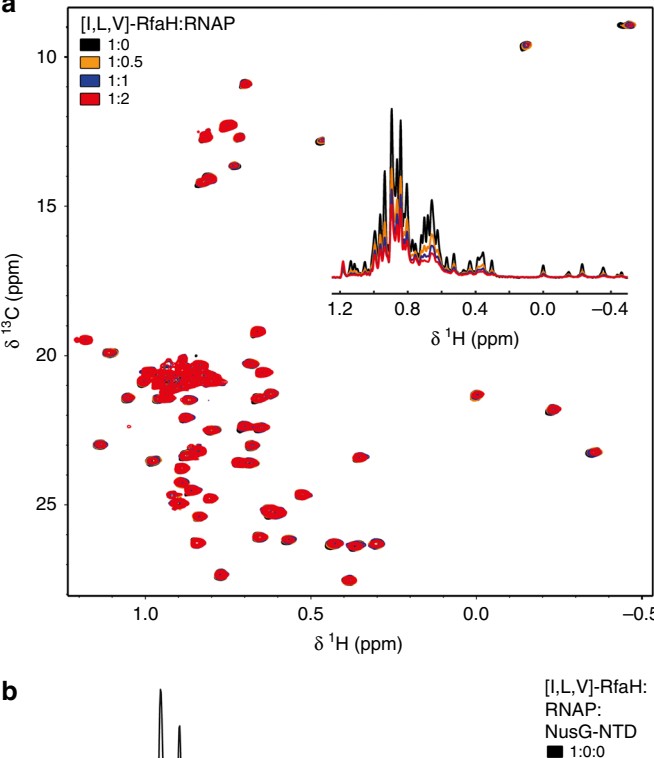

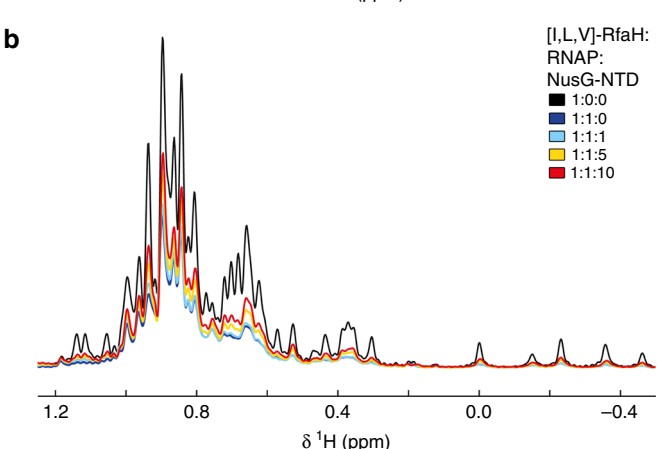

**Fig. 3** Binding of RfaH to RNAP. **a** 2D [¹H, ¹³C] methyl-TROSY spectra of 15 μM [I,L,V]-RfaH titrated with RNAP (concentration of stock solution: 78 μM). Inset: normalized 1D [¹H, ¹³C]-methyl-TROSY spectra, colored as 2D spectra. See also Supplementary Figure 2. **b** Displacement of RfaH from RNAP by NusG-NTD. Normalized 1D methyl-TROSY spectra of [I,L,V]-RfaH (30 μM, black), [I,L,V]-RfaH in the presence of equimolar RNAP (30 μM each, red), and [I,L,V]-RfaH upon titration of the [I,L,V]-RfaH:RNAP complex (30 μM each) with 712 μM NusG-NTD; molar ratio of [I,L,V]-RfaH: RNAP:NusG-NTD is indicated in color

CTD in the absence and presence of *ops*EC (Supplementary Figure 5a–e). The diffusion coefficient of [I,L,V]-RfaH:*ops*EC is significantly smaller than that of free [I,L,V]-RfaH or [I,L,V]-RfaH-CTD (Supplementary Table 1), confirming that the all-β signals in the [I,L,V]-RfaH:*ops*EC sample arise from RfaH bound to *ops*EC and that signals of the freed RfaH-CTD are visible even when RfaH is bound to the *ops*EC.

The decrease of RfaH-NTD methyl group signal intensity likely is a combination of two effects: a general decrease resulting from the increased molecular mass of [I,L,V]-RfaH upon complex formation and a non-uniform decrease due to slow or intermediate exchange on the chemical shift timescale. Thus we analyzed the signal intensity in certain titration steps quantitatively to identify affected residues as established[39] (Fig. 4b). In brief, in each titration step, relative intensity, i.e., the ratio of

remaining signal intensity to that in the spectrum of free [I,L,V]-RfaH, was determined. Residues with relative signal intensities below certain thresholds were classified as either strongly or moderately affected (for details see Methods section). Mapping of the relative signal intensity of the 1:0.5 complex on the three-dimensional structure of RfaH-NTD revealed a patch where signal intensity changed significantly (Fig. 4c). To aid visualization limited by a small number of NMR-active probes, we graphically extended the representation of affected residues by including the two flanking residues on each side, unless they were an unaffected Ile, Leu, or Val residue (beige in Fig. 4c). Comparing the affected regions with the cryo EM structure of the RfaH:*ops*EC[13] shows that the main, high-affinity contacts with the β′CH are in good agreement (Fig. 4d). The HTTT motif in helix α2 of RfaH interacts with the βGL[13,26], but since this motif lacks NMR-active probes and the closest labeled residues point to the interior of RfaH, no information on these contacts could be obtained (Fig. 4c). These results demonstrate that in the presence of *ops*EC RfaH domains dissociate, RfaH-NTD binds to the EC, and RfaH-CTD refolds into the β-state, confirming that the *ops*-paused EC is the relevant signal for RfaH recruitment.

**EC-bound RfaH interacts with S10**. In a subpopulation of the RfaH:*ops*EC complexes observed by cryo EM, the RfaH-CTD binds to the β-flap tip helix at the RNA exit channel[13]. To test if this interaction also occurs in solution, we performed a titration of [I,L,V]-RfaH-CTD with *ops*EC (Supplementary Figure 5f). In the 1D methyl-TROSY spectra signal intensity of [I,L,V]-RfaH-CTD decreases by ~25% upon addition of *ops*EC. This loss of intensity indicates complex formation as the molecular mass of [I,L,V]-RfaH-CTD increases upon *ops*EC binding, although this interaction might be weak. These observations are in agreement with the finding that the diffusion coefficient of [I,L,V]-RfaH-CTD is slightly decreased in the presence of *ops*EC (Supplementary Table 1). The finding that the signals of the all-β RfaH-CTD, i.e., the freed CTD, are visible when RfaH is bound to the *ops*EC strengthens the hypothesis that RfaH-CTD is only transiently bound to RNAP in the EC.

We argued that RfaH recruits a ribosome via interactions observed in a binary complex of isolated RfaH-CTD and S10[31]. To test if this contact is preserved when RfaH is bound to the *ops*EC, we performed an NMR-based titration of [I,L,V]-RfaH with S10 in the presence of the *ops*EC using S10 lacking the ribosome-binding loop (S10Δ) in complex with NusB to increase stability[40]. Upon addition of protonated *ops*EC to [I,L,V]-RfaH in equimolar concentration, mainly signals of the β-barrel CTD were observable, showing that RfaH is bound to the *ops*EC and that the CTD is in the all-β state (Fig. 5a). Subsequent titration with S10Δ:NusB decreased intensity of some of these signals significantly (Fig. 5a, b). Affected residues are located in β-strands 3 and 4 as well as in the connecting loop (Fig. 5c), in agreement with the binding site observed in the binary RfaH-CTD:S10Δ complex (Fig. 5d and Supplementary Figure 6). Thus, the S10 interaction site of RfaH-CTD is accessible in the *ops*EC:RfaH complex, consistent with the cryo EM structure of the RfaH: *ops*EC complex[13] and our data that RfaH-CTD:S10 interaction is required for translation activation[31].

**RfaH is recycled upon release from the EC**. The presence of RfaH-NTD is sufficient to induce the RfaH-CTD folding into an α-state that is energetically unfavorable in the isolated domain[31,32,41], leading us to propose that RfaH transforms back into the autoinhibited state after the EC dissociates at a terminator[32]. Testing this hypothesis at a canonical terminator by NMR spectroscopy would be challenging because such a complex

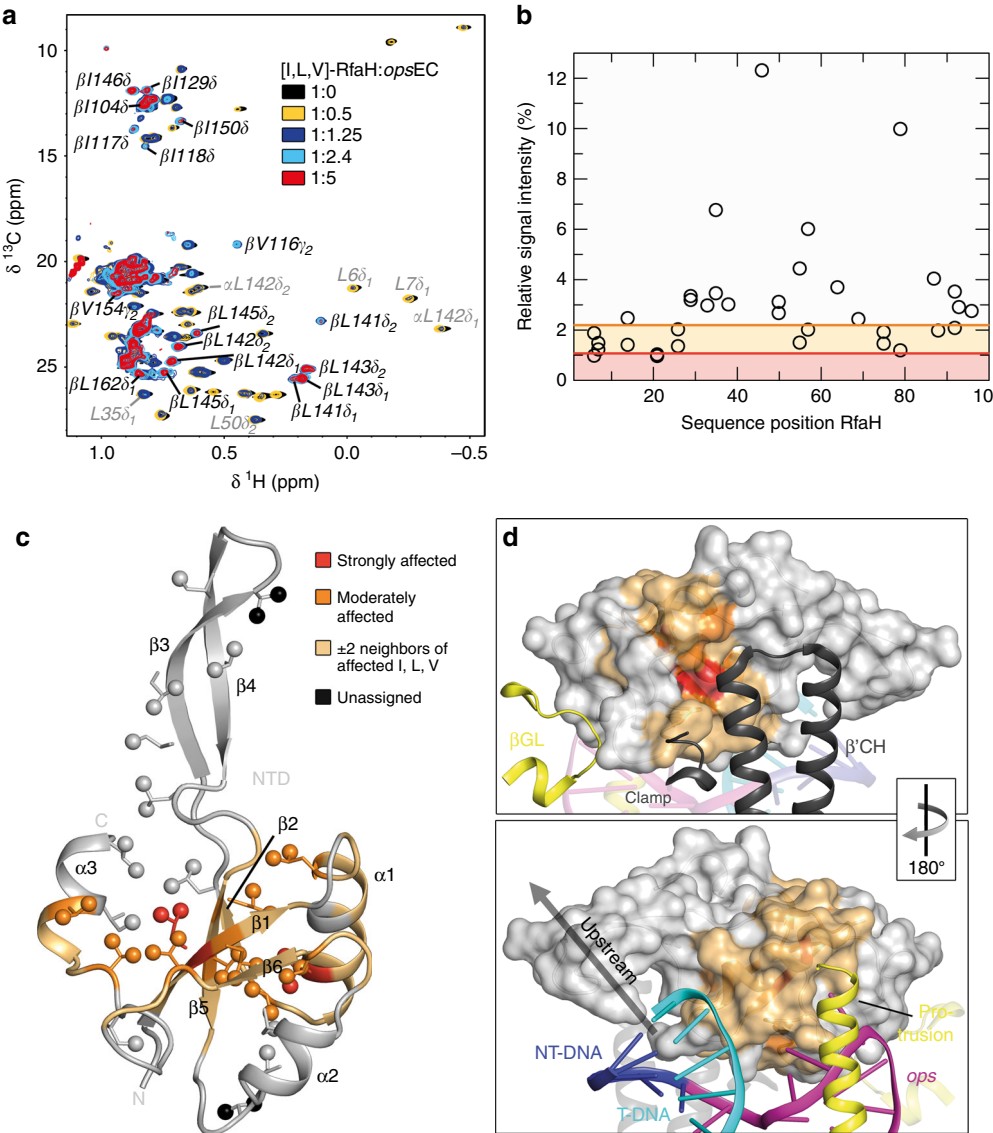

**Fig. 4** RfaH recruitment to the *ops*EC. **a** 2D [$^1$H, $^{13}$C] methyl-TROSY spectra of [I,L,V]-RfaH in the absence or presence of *ops*EC assembled with $^2$H-RNAP (concentration of [I,L,V]-RfaH: 233 μM (1:0), 54 μM (1:0.5), 27 μM (1:1.25), 15 μM (1:2.4), 8 μM (1:5)). α and β indicate the all-α or all-β state of the RfaH-CTD. **b** Relative signal intensity of [I,L,V]-RfaH-NTD methyl groups with 0.5 equivalents of *ops*EC. Orange and red lines indicate thresholds for moderately (60% of average relative intensity) and strongly (30% of average relative intensity) affected methyl groups, respectively. Source data are provided as a Source Data file. **c** Mapping of affected methyl groups onto RfaH-NTD structure (ribbon representation; light gray; PDB ID: '5OND'). Ile, Leu, and Val residues are in stick representation with the carbon atom of the methyl groups as sphere. Termini and secondary structure elements are labeled. The representation was graphically extended by including the two flanking residues on each side of an affected residue (beige) as established[39]. **d** RfaH-NTD bound to the *ops*EC (PDB ID: '6C6S'). RfaH-NTD is in surface representation, color code as in **c**, DNA and selected elements of the RNAP are in ribbon representation and labeled. The arrow indicates how the structures are rotated with respect to each other

is unstable. Instead, we induced [I,L,V]-RfaH release from the *ops*EC by addition of a 10-fold molar excess of NusG-NTD and monitored RfaH displacement by recording methyl-TROSY spectra (Fig. 6a). The addition of protonated *ops*EC to [I,L,V]-RfaH in a 1:1 molar ratio led to the disappearance of signals corresponding to autoinhibited RfaH and mainly β-barrel CTD signals were observable, confirming RfaH recruitment and transformation. Upon titration of [I,L,V]-RfaH:*ops*EC with pro-tonated NusG-NTD, all-β CTD signals were partially replaced by signals of autoinhibited RfaH (Fig. 6a), consistent with RfaH displacement from the *ops*EC followed by recycling into its autoinhibited state.

We next wanted to probe the fate of RfaH released from RNAP in a more natural pathway, upon completion of RNA synthesis.

The autoinhibited RfaH depends on wild-type (WT) *ops* site for recruitment and cannot act on a G8C *ops* template where the NT-DNA hairpin is disrupted[29]. By contrast, the isolated RfaH-NTD can bind to the EC at any site[30] and we showed that the RfaH-NTD as well as RfaH variants locked in the open state due to substitutions at the NTD-CTD interface are recruited to RNAP transcribing the G8C template[33]. Here we used a two-step in vitro assay (Fig. 6b) to test if released RfaH regains its autoinhibited state, and thus dependence on *ops* for recruitment. In the first step, a linear DNA template containing T7A1 promoter and the *ops* element was immobilized on streptavidin beads via a biotin moiety. Transcription was carried out by *E. coli* RNAP in the presence of full-length RfaH (RfaH$^{FL}$) and the supernatant containing released RfaH (RfaH$^{SN}$) was collected. In the second

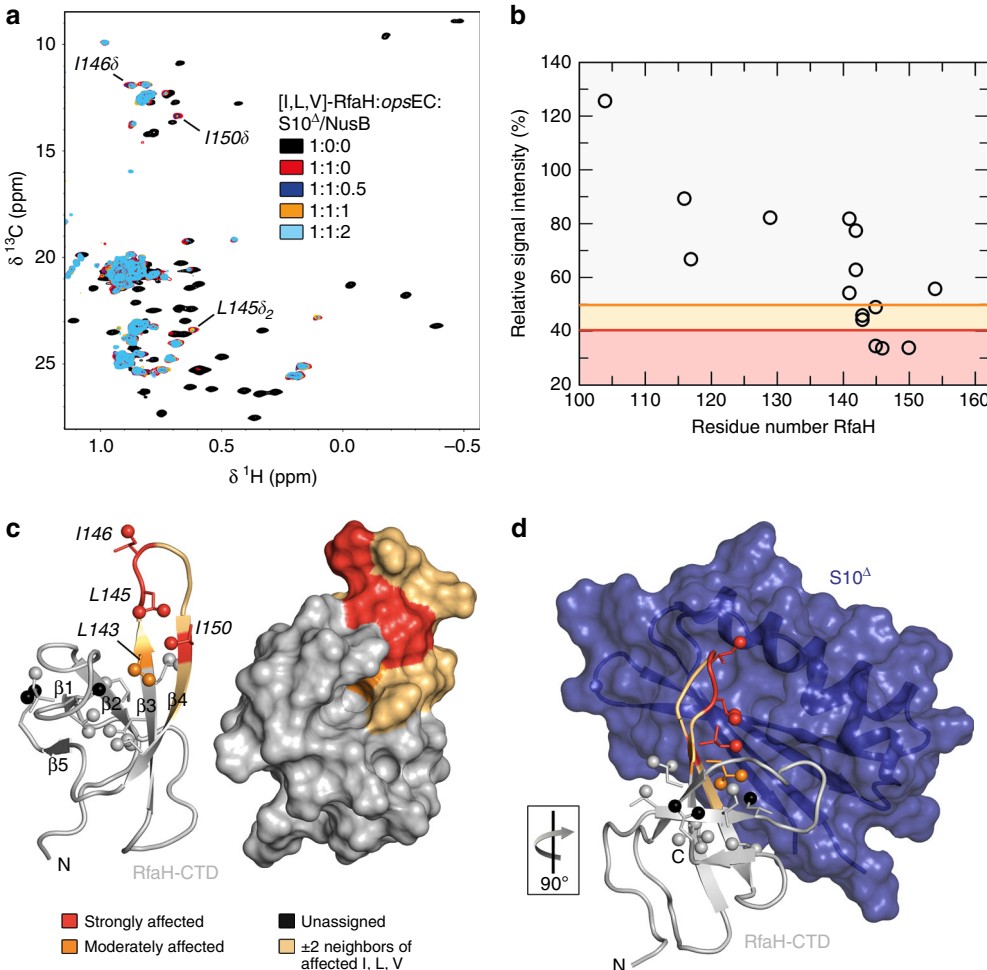

**Fig. 5** Structural basis of translation activation by RfaH. **a** 2D [$^1$H, $^{13}$C] methyl-TROSY spectra of [I,L,V]-RfaH alone (200 μM), in the presence of equimolar concentration of *ops*EC (23 μM), and upon titration of RfaH:*ops*EC with 234 μM S10$^\Delta$:NusB; molar ratio [I,L,V]-RfaH:*ops*EC:S10$^\Delta$:NusB is indicated in color. Resonances with significant intensity changes are labeled. **b** Methyl-TROSY-derived relative intensity of [I,L,V]-RfaH methyl groups after addition of one equivalent of *ops*EC and two equivalents of S10$^\Delta$:NusB vs. sequence position in RfaH. Orange and red lines indicate thresholds for moderately affected (80% of the average relative intensity) and strongly affected (65% of the average relative intensity) methyl groups, respectively. Source data are provided as a Source Data file. **c** Mapping of affected methyl groups onto RfaH-CTD structure (PDB ID: '2LCL'). RfaH (gray) is shown in ribbon (left) and surface (right) representation, methyl groups are shown as spheres and are color-coded. **d** Model of the RfaH-CTD:S10$^\Delta$ complex based on the NusG-CTD:S10$^\Delta$ complex (PDB ID: '3D3B'). S10$^\Delta$ in ribbon and surface representation (blue), representation of RfaH-CTD as in **c**. The orientation of RfaH-CTD relative to **c** is indicated

step, RfaH$^{SN}$ was added to halted radiolabeled ECs formed on templates with either WT or G8C *ops*. Following the addition of NTP substrates, the RNA products collected at different times were analyzed by gel electrophoresis (Fig. 6c) and quantified.

On the WT *ops* template, RfaH$^{FL}$ reduced RNAP pausing at U38 ~4-fold and delayed RNAP escape from the *ops* site (G39 + C40 positions) ~4-fold (Fig. 6d). RfaH-NTD and RfaH$^{SN}$ had very similar effects. A control in which RNAP release was prevented by a protein roadblock (RB; see Methods section) demonstrated that under these conditions all RfaH was bound to RNAP, as no activity was present in the supernatant. Notably, at low GTP (5 μM) used in these experiments to enable manual sampling, RfaH-induced pause at G39 + C40 masks its antipausing effects downstream, and the run-off transcript yields do not increase in the presence of RfaH. As expected, RfaH-NTD stimulated productive RNA synthesis on the G8C *ops* template ~2.5-fold, whereas neither RfaH$^{FL}$ nor RfaH$^{SN}$ had any effect. These results suggest that RfaH regains the autoinhibited, *ops*-dependent state after the EC dissociates at the end of the linear

DNA template, in support of the direct observation of reverse transformation by NMR spectroscopy (Fig. 6a).

## Discussion

The results presented here support our earlier hypothesis that RfaH operates in a true cycle, which begins and ends with the inactive, autoinhibited state (Fig. 7). RfaH recruitment is unusually complex, with the *ops*EC serving as a minimal signal (Fig. 4); while RfaH can weakly interact with the *ops*DNA[29] or core RNAP (Fig. 2), it binds to the *ops*EC with ~1000-fold higher affinity that matches its cellular concentrations[38]. When RNAP pauses at the *ops* site, the NT strand forms a hairpin, exposing the nucleotides in the loop region at the enzyme's surface, allowing sequence-specific recognition by RfaH[13,29]. The delay of RNAP at the *ops* site is thought to provide a crucial time window during which autoinhibited RfaH locates its few genomic targets and establishes interactions with certain RNAP elements (likely the βGL) and the accessible *ops* nucleotides, stabilizing the NT-DNA

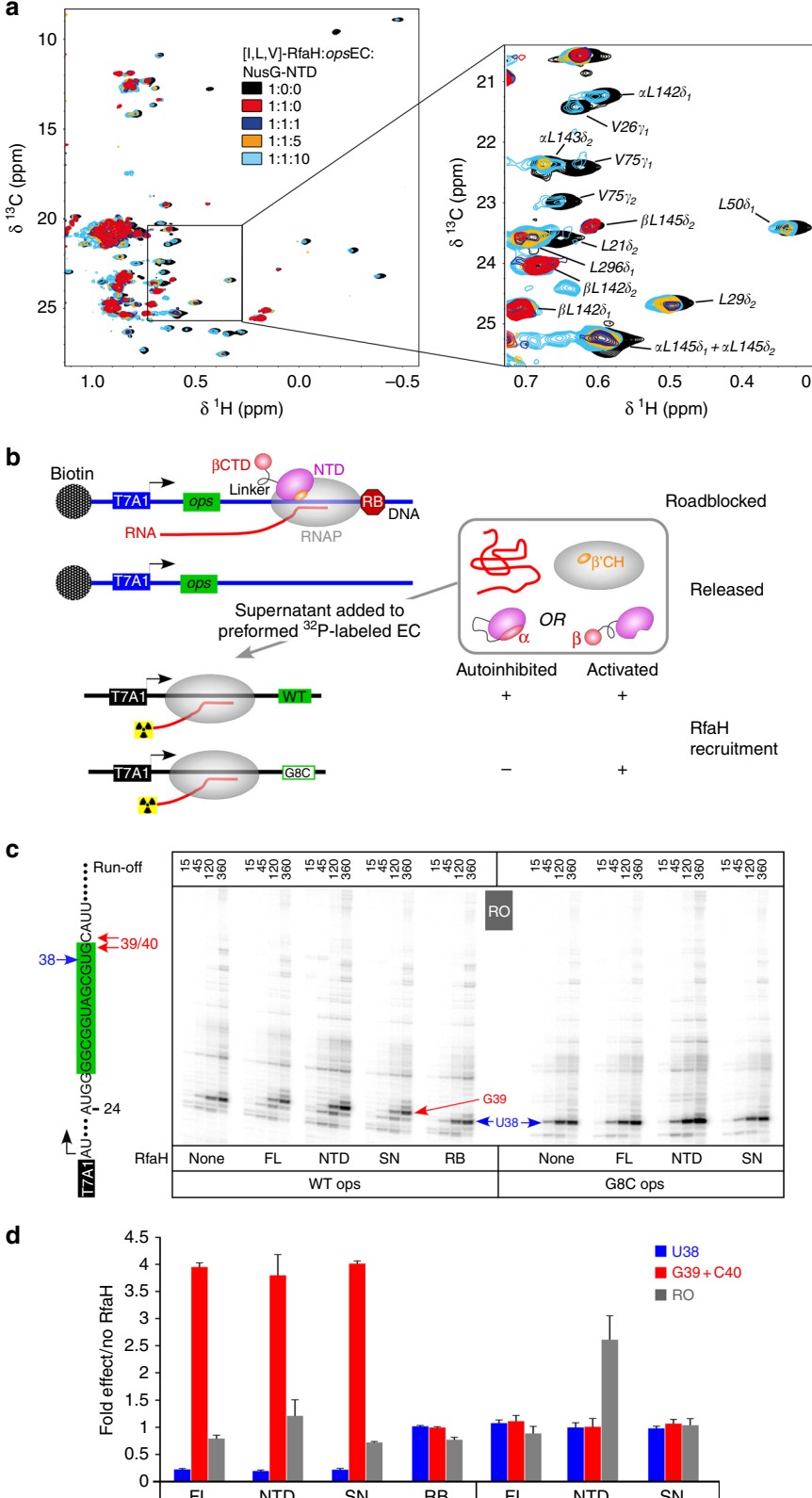

hairpin and forming a transient encounter complex. In this complex, RfaH is positioned near its final binding site on the β′ CH, but the high-affinity NTD:β′CH contacts are precluded by the α-helical CTD. As the autoinhibited state of RfaH does not exchange with an open conformation on the NMR timescale, the functional role of the encounter complex remains to be determined, although several possibilities are conceivable. (i) Contacts

in the encounter complex could pre-orient RfaH and increase its local concentration near the β′CH, facilitating RfaH-NTD binding to the tip of the β′CH. (ii) The encounter complex could induce conformational changes that destabilize the interdomain interface and ultimately lead to transient domain opening. Although not being observable in our experiments, we cannot rule out that the binding of RfaH to *ops*DNA or RNAP alone may

**Fig. 6** Recycling of RfaH. **a** 2D [¹H, ¹³C] methyl-TROSY spectra of [I,L,V]-RfaH alone (200 μM), in the presence of equimolar concentration of *ops*EC (23 μM), and upon titration of RfaH:*ops*EC with NusG-NTD (concentration of stock solutions 240 μM and 486 μM); molar ratio [I,L,V]-RfaH:*ops*EC:NusG-NTD is indicated in color. α and β indicate the all-α or all-β state of the RfaH-CTD. **b** Experimental set-up to follow RfaH state using in vitro transcription assay. **c** Determination of RfaH effect on single-round RNA synthesis. The relevant RNA region is shown on the left, with the *ops* element highlighted in green. Prominent pause sites (U38, G39, and C40) are indicated. Halted α³²P-labeled A24 ECs were chased in the presence of RfaH-NTD, RfaH^FL, or supernatants from roadblocked (RB) or free (SN) first-round reactions on the WT or G35C (corresponds to G8C in the *ops* element) template. Reactions were quenched at the indicated times (in seconds) and analyzed on 10% denaturing acrylamide gels; a representative gel is shown. **d** The fractions of RNA species indicated were determined from 360-s time points. The ratios of RNA in the presence and in the absence of the RfaH variant indicated were determined from three independent biological replicates and are shown as mean ± standard deviation. Source data are provided as a Source Data file

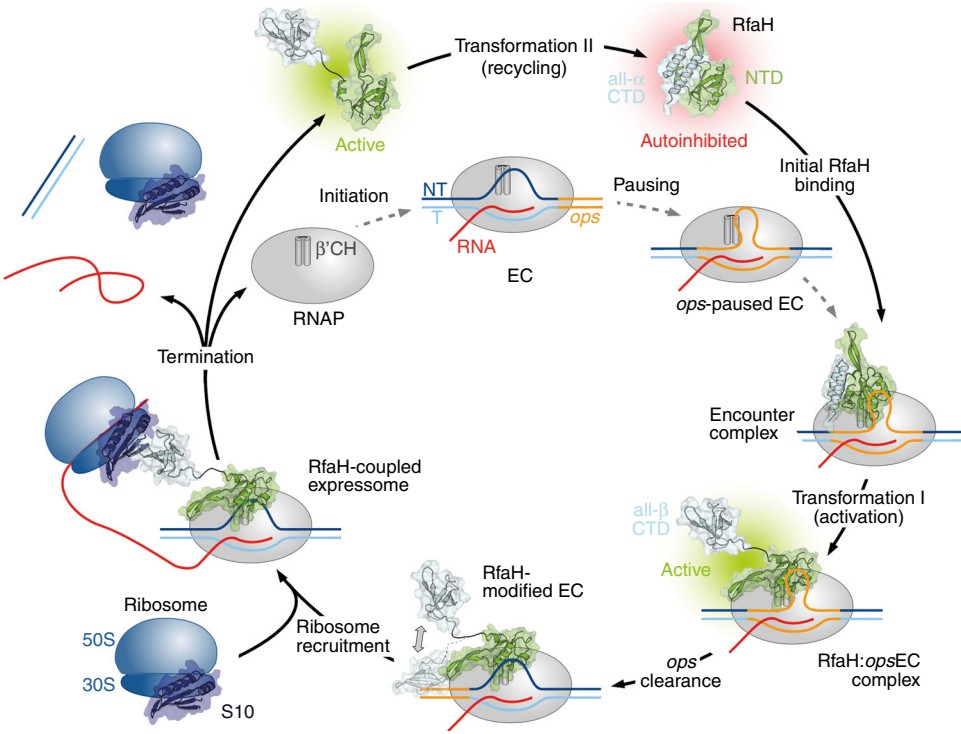

**Fig. 7** Functional cycle of RfaH. Structural transformations of the interdomain interface and the RfaH-CTD underlie reversible switching between the autoinhibited and active states of RfaH

cause such changes in the RfaH-NTD:RfaH-CTD interface. However, as only the central *ops* bases are exposed on the surface of the RNAP[13], RfaH will inevitably interact with certain RNAP elements as soon as it establishes contacts with the *ops* hairpin. Thus, we hypothesize that a combination of interactions with *ops*DNA and RNAP provokes the weakening of the RfaH domain interface. Although we do not observe either scenario, we favor a model where elements of both mechanisms underlie RfaH activation. Upon dissociation of the RfaH domains the encounter complex is converted into a stable RfaH:EC complex. The RfaH-NTD maintains interactions with the EC throughout elongation, increasing transcription processivity through stabilizing contacts with the NT-DNA and the upstream duplex DNA, as well as by blocking RNAP swiveling that occurs during pausing[13,42]. The released RfaH-CTD transforms and binds S10 (Fig. 5), converting RfaH into a potent activator of translation initiation[31] and possibly coupling transcription to translation elongation, as proposed for NusG[18]. Finally, RfaH completes the cycle by transforming back into the autoinhibited state upon release from RNAP (Fig. 6).

Observations that all NusG homologs promote productive transcription, with the NTD being sufficient for this activity[30,43], led us and others to focus on the NTD-dependent modification of

RNAP. Yet the regulatory diversity of NusG homologs is conferred by their CTDs, which interact with different partners to ensure coordination between RNA synthesis and posttranscriptional events. Comparison of RfaH- and NusG-CTDs reveals a combination of similar and distinct activities. Both CTDs interact similarly with S10[5,31], suggesting that they may bridge RNAP and the lead ribosome; the available evidence is consistent with RfaH recruitment of ribosome[31] and with coupling by NusG[18] but a systematic analysis remains to be done. In contrast, other interactions/roles are different. First, NusG-CTD binds to Rho, promoting termination at suboptimal sequences through favoring the closed, translocation-competent ring state[6,7], whereas RfaH does not bind to Rho and inhibits Rho-dependent termination[26]. Second, RfaH-CTD could affect folding of the nascent RNA through transient contacts to the RNAP exit channel[13], although the significance of this interaction remains to be determined. Finally, the RfaH-CTD prevents off-target recruitment and thus competition with NusG through autoinhibition mediated by transformation, maintaining the separation of the RfaH and NusG regulons.

Even though RfaH and NusG primary sequences are quite divergent[33], only a few residues determine their key regulatory differences. Activation of Rho-dependent termination is

determined by a 5-residue surface loop of NusG that, when grafted onto the RfaH-CTD, is sufficient for stimulation of Rho transition into the active conformation[6]. Similarly, the existence of the autoinhibited state is controlled by a few residues at the interface between the RfaH-NTD and RfaH-CTD; a single substitution of each of two key RfaH residues for their NusG counterparts disrupts the interface and alleviates the requirement for *ops* during recruitment[33]. In contrast, RfaH binding to the NT-DNA relies on readout of the primary sequence and the secondary structure of the *ops* hairpin by many RfaH residues, which are not conserved in NusG[13,29].

Autoinhibition is a widespread regulatory mechanism in which intramolecular interactions between separate regions of a polypeptide negatively regulate its function, allowing temporal and spatial regulation of cellular processes by limiting activation to certain physiological conditions[44]. Autoinhibition modulates function of diverse proteins, from transcription factors[45] to protein kinases[46], and is implicated in virulence[47] and disease[48]. Autoinhibition enables tight regulation, particularly when thousands of potential targets need to be distinguished, as is the case with E3 ubiquitin ligases[49].

NusG cooperates with Rho to promote termination at suboptimal sites[17], an essential function of NusG to silence foreign DNA[50]. Although being outnumbered by NusG 100:1[51], RfaH efficiently outcompetes NusG for binding to the EC[32] and abolishes Rho-mediated termination[26]. To prevent interference with essential NusG, RfaH recruitment must be strictly limited to *ops* operons, making attainment of autoinhibition a key step in the specialization of RfaH. Contrasting other cases of autoinhibition, autoinhibition in RfaH is achieved not only by the interaction of two domains, but is coupled to the transformation of a whole domain into a conformation, the all-α form, that does not correspond to the structure of the isolated domain[31,32].

It is not known whether other specialized NusG paralogs are autoinhibited and, if so, how they are activated and whether the CTD refolds similarly. In RfaH, transient contacts to the *ops*DNA hairpin and flanking RNAP regions are required to trigger domain dissociation, maybe via an encounter complex. While autoinhibited RfaH-like proteins could use analogous contacts to the NT-DNA strand and the paused EC for activation, other mechanisms could be envisioned, such as allosteric activation; e.g., small ligands could bind to either domain to weaken their interactions.

While in *E. coli* NusG, the domains move independently, and no intramolecular domain interactions can be detected[52], autoinhibition has been observed for *Thermotoga maritima* NusG[53]. Here, in contrast to RfaH, autoinhibition is accomplished by the β-barrel CTD, which shields the RNAP-binding site on the NTD and most probably provides thermal stabilization[54]. Why does RfaH use an α-helical hairpin?

To answer this question, we generated a model of RfaH where the all-β CTD interacts with the NTD as in *Tm*NusG (RfaH[βCTD]; Supplementary Figure 7). The linker is sufficiently long and RfaH[βCTD] can be easily integrated into a closed conformation without steric problems. Analysis of the domain interactions reveals that in RfaH the binding surface is larger than in RfaH[βCTD] (~900 Å² vs. ~700 Å²), resulting in a significantly more negative solvation energy (~−14 kcal/mol vs. −8 kcal/mol). The stronger domain interaction in RfaH may be required to prevent an equilibrium between open and closed state, consistent with our present results, and thus uncontrolled activation.

We speculate that a need to tightly control the off-target recruitment necessitates the transformation-coupled autoinhibition in RfaH, especially as the use of the all-α CTD state imparts dual autoinhibition—the closed state lacks the binding sites for both RNAP and the ribosome, potentially further minimizing deleterious effects of spurious RfaH activation. Studies of structures and recruitment of NusG paralogs from other species will reveal their underlying specificity mechanisms.

The thermodynamic hypothesis states that, under physiological conditions, a protein adopts the three-dimensional structure which corresponds to the state of the lowest Gibbs free energy for the whole system. This so-called physiological state is solely determined by the totality of interactions and thus the amino acid sequence[55]. Most proteins exist as an ensemble of closely related equilibrium structures in their energetically stable state and thus follow this one-sequence, one-fold paradigm. However, more and more chameleonic/metamorphic proteins[56] that defy this paradigm are found [see ref. [57] for a recent review]. In these metamorphic proteins, more than one distinct structural form is energetically favored.

In most cases, the metamorphic regions are small (5–14 residues) and metamorphosis involves either transitions between unstructured to structured states or conformational switching (α ↔ β). Transitions can be triggered by simple external cues (e.g., pH, temperature, salt concentration; refs. [58,59]) may be driven by evolutionary pressure as observed in the Cro family of bacteriophage transcription factors[60], or may underlie biological activity, such as regulation of chemotaxis by lymphotactin[61], pore formation by lytic toxins[62], regulation of the circadian clock by KaiB[63], or photoreactivation in cytochromes[64]. As in classical metamorphoses, the observed changes are usually unidirectional, although reversible refolding has been reported[61,63,65].

What sets RfaH apart from other metamorphic proteins are (i) the scale of the reversible transformation, in which the entire 50-residue domain refolds, (ii) distinct and essential biological functions of both alternative folds, and (iii) the fact that this dramatic behavior occurs in a member of the only universally conserved family of transcription factors. The fold of the CTD is solely determined by the presence or absence of the NTD[32], i.e., the information which fold to adopt is encoded in RfaH itself. A major determinant of the NTD:CTD interaction is the salt bridge E48:R138 as its elimination leads to a coexistence of the autoinhibited state and the open form with the CTD in the β-barrel conformation[31], turning RfaH into a NusG-like general transcription factor as the dependence on *ops* is abolished[31]. Arguments (i) and (ii) prompted us to name RfaH a transformer protein[34]. An α→β switch of a whole protein/domain is only known for amyloidogenic proteins, such as prions[66], but the two states cannot coexist, the transition is irreversible, and the resulting β-aggregates are pathogenic.

In conclusion, metamorphosis is an increasingly recognized regulatory tool in nature, but the functional and conformational plasticity coupled with autoinhibition of RfaH sets new standards for regulation and suggests that similar principles are exploited by many transformer proteins awaiting discovery.

## Methods

**Cloning**. The gene encoding RfaH was amplified from pIA238 using primers Fw_rfaH_pET19bmod and Rv_rfaH_pET19bmod (Supplementary Table 2) and cloned into pET19bmod, a variant of pET19b, via NdeI and *Bam*HI restriction sites. The recombinant target protein carries a hexahistidine (His₆) tag followed by a tobacco etch virus (TEV) protease cleavage site at its N-terminus.

**Gene expression and protein purification**. All *E. coli* strains used in gene expression were derivatives of *E. coli* B and grown at temperatures between 16 and 37 °C. Antibiotics were added to the medium as follows: ampicillin 100 μg/ml, carbenicillin 100 μg/ml, kanamycin 50 μg/ml, and chloramphenicol 34 μg/ml. The source organisms for all proteins used in this work are derivatives of *E. coli* K. All expression plasmids are listed in Supplementary Table 3.

RfaH for NMR studies was produced as described in ref. [31]. In brief, *E. coli* BL21 (λ DE3) cells (Novagen, Madison, WI, USA) harboring plasmid pET19bmod_RfaH were grown in lysogeny broth (LB) medium supplemented with kanamycin to an optical density at 600 nm ($OD_{600}$) of 0.6 at 37 °C. The temperature was lowered to

20 °C and gene expression was induced after 30 min by addition of 0.2 mM isopropyl-1-thio-β-D-galactopyranoside (IPTG). Cells were harvested after overnight incubation by centrifugation (6000 × g), resuspended in buffer A$^{RfaH}$ (50 mM tris(hydroxymethyl)aminomethane (Tris)/HCl (pH 7.5), 300 mM NaCl, 5% (v/v) glycerol, 1 mM dithiothreitol (DTT)) supplemented with 10 mM imidazole, DNase I (AppliChem GmbH, Darmstadt, Germany), and 1/2 protease inhibitor tablet (cOmplete, EDTA-free, Roche Diagnostics, Mannheim, Germany) and lysed using a microfluidizer. The lysate was cleared by centrifugation and the soluble fraction was then applied to a HisTrap column (column volume 1–5 ml, GE Healthcare, Munich, Germany) that was subsequently washed with buffer A$^{RfaH}$ supplemented with 10 mM imidazole. A step gradient from 100 mM to 1 M imidazole in buffer A$^{RfaH}$ was used for elution. RfaH-containing fractions were combined and dialyzed against buffer B$^{RfaH}$ (50 mM Tris/HCl (pH 7.5), 150 mM NaCl, 5% (v/v) glycerol, 1 mM DTT). After 2 h TEV protease was added and cleavage was carried out overnight at 4 °C. The solution was then again applied on a 5 ml HisTrap column (GE Healthcare, Munich, Germany). The target protein was collected in the flow-through, concentrated by ultrafiltration, flash-frozen in liquid nitrogen, and stored at −80 °C.

RfaH for transcription assays was produced similarly, except that plasmid pIA238 was used for the expression, resulting in E. coli RfaH with N-terminal His$_6$-tag followed by a thrombin cleavage site. Thus, cleavage was carried out during overnight dialysis at room temperature in the presence of thrombin instead of TEV protease.

The production of RfaH-CTD was according to ref. [31] and the conditions were similar to the ones used for full-length RfaH. For expression E. coli BL21 (DE3) cells containing pETGB1a_EcrfaH-CTD(101-162) were used. The plasmid codes for E. coli RfaH-CTD with N-terminal His$_6$-Gb1 tag followed by a TEV protease cleavage site. For purification a 5 ml Ni$^{2+}$-HiTrap column (GE Healthcare, Munich, Germany) was used and buffer A$^{RfaH-CTD}$ consisted of 50 mM Tris/HCl (pH 7.5), 150 mM NaCl. The pure target protein was finally dialyzed against 25 mM 4-(2-hydroxyethyl)-1-piperazineethanesulfonic acid (HEPES; pH 7.5), 100 mM NaCl, concentrated by ultrafiltration, flash-frozen in liquid nitrogen, and stored at −80 °C.

For the production of NusG-NTD[52] E. coli BL21 (λ DE3) cells harboring plasmid pET11a_EcNusG-NTD(1-124), which encodes E. coli NusG 1-124, were grown in ampicillin-containing LB medium to an $OD_{600}$ of 0.8 at 37 °C. Overexpression was induced by addition of 1 mM IPTG. After 4 h cells were harvested by centrifugation (6000 × g), resuspended in buffer A$^{NusG-NTD}$ (50 mM Tris/HCl (pH 7.5), 150 mM NaCl) supplemented with DNase I (AppliChem GmbH, Darmstadt, Germany) and 1/4 protease inhibitor tablet (cOmplete, EDTA-free, Roche Diagnostics, Mannheim, Germany), and lysed with a microfluidizer. After centrifugation, nucleic acids were precipitated by addition of streptomycine sulfate (1% (w/v)). Upon centrifugation (NH$_4$)$_2$SO$_4$ was added to the supernatant to a concentration of 50% (w/v), precipitating NusG-NTD. The precipitate was pelleted by centrifugation and dissolved in buffer B$^{NusG-NTD}$ (10 mM Tris/HCl (pH 7.5)). The solution was dialyzed against buffer B$^{NusG-NTD}$ before being applied to a 5 ml HeparinFF column (GE Healthcare, Munich, Germany). The column was washed with buffer B$^{NusG-NTD}$ and the target protein was eluted by a NaCl step gradient from 50 mM to 1 M in buffer B$^{NusG-NTD}$. NusG-NTD containing fractions were combined, concentrated by ultrafiltration, and applied to a HiLoad S75 size exclusion column (GE Healthcare, Munich, Germany) equilibrated with buffer C$^{NusG-NTD}$ (25 mM HEPES (pH 7.5), 100 mM NaCl). NusG-NTD containing fractions were combined and the solution was concentrated by ultrafiltration and flash-frozen in liquid nitrogen before being stored at −80 °C.

The production of S10$^Δ$:NusB was based on ref. [67]. Briefly, E. coli BL21 (λ DE3) cells harboring the plasmids for either S10$^Δ$ (pGEX-6P_ecoNusE$^Δ$; encodes E. coli S10$^Δ$ with N-terminal glutathione S-transferase (GST)-tag followed by PreScission protease cleavage site) or NusB (pET29b_ecoNusB; encodes E. coli NusB), were grown in LB medium containing ampicillin or kanamycin, respectively, at 37 °C to an $OD_{600}$ of 0.5. The temperature was lowered to 20 °C and gene expression was induced after 30 min by addition of 0.5 mM IPTG. After overnight incubation cells were harvested by centrifugation (6000 × g). Cell pellets of S10$^Δ$ and NusB-containing cells, obtained from the same culture volume, were resuspended in buffer A$^{S10Δ:NusB}$ (50 mM Tris/HCl (pH 7.5), 150 mM NaCl, 1 mM DTT) and combined. Cells were subsequently lysed using a microfluidizer and the lysate was stirred for 30 min at 4 °C to ensure formation of the S10$^Δ$:NusE complex. The extract was then cleared by centrifugation and applied to four coupled 5 ml GSTrap FF columns (GE Healthcare, Munich, Germany) equilibrated with buffer A$^{S10Δ:NusB}$. After washing with buffer A$^{S10Δ:NusB}$ the complex was eluted with buffer A$^{S10Δ:NusB}$ containing 15 mM reduced glutathione. The S10$^Δ$:NusB solution was supplemented with PreScission protease and dialyzed against buffer B$^{S10Δ:NusB}$ (50 mM Tris/HCl (pH 7.5), 1 mM DTT) overnight. The protein solution was applied to two 5 ml HiTrap Q XL columns (GE Healthcare, Munich, Germany) coupled to two HiTrap SP XL columns (GE Healthcare, Munich, Germany). Upon washing with buffer B$^{S10Δ:NusB}$ the HiTrap SP XL columns were disconnected and S10$^Δ$:NusB were eluted with buffer B$^{S10Δ:NusB}$ containing 1 M NaCl. The solution was dialyzed against 25 mM HEPES (pH 7.5), 100 mM NaCl, concentrated by ultrafiltration, before being flash-frozen in liquid nitrogen, and stored at −80 °C.

RNAP for in vitro transcription assays was produced according to ref. [68]. E. coli BL21 (λ DE3) cells harboring pVS10 (encoding E. coli RNAP subunits α, β, β′ with C-terminal His$_6$ tag, and ω) were grown at 37 °C in carbenicillin-containing LB medium to an $OD_{600}$ of 0.75 before overexpression was induced by 1 mM IPTG for

3 h. Cells were harvested by centrifugation (6000 × g) and resuspended in buffer A$^{RNAP1}$ (50 mM Tris/HCl (pH 6.9), 500 mM NaCl, 5% (v/v) glycerol) supplemented with one protease inhibitors cocktail (Roche Applied Science) and 1 mg/ml lysozyme. Cell lysis was carried out by sonication and the cleared extract was supplemented with 20 mM imidazole before being loaded onto a His GraviTrap column (GE Healthcare Life Science). The column was washed with buffer A$^{RNAP1}$ containing 20 mM imidazole and RNAP was eluted with buffer A$^{RNAP1}$ containing 250 mM imidazole. The protein solution was dialyzed against buffer B$^{RNAP1}$ (50 mM Tris/HCl (pH 6.9), 5% (v/v) glycerol, 0.5 mM EDTA, 1 mM DTT) supplemented with 75 mM NaCl and was then applied to a HiPrep Heparin FF column (GE Healthcare Life Science) to remove nucleic acids. The column was washed with buffer B$^{RNAP1}$ containing 75 mM NaCl and RNAP was eluted with a constant NaCl gradient from 75 mM to 1.5 M in buffer B$^{RNAP1}$. Target protein containing fractions were dialyzed against buffer B$^{RNAP1}$ containing 75 mM NaCl and applied on a MonoQ column (GE Healthcare Life Science). Washing and elution were analogous to the Heparin affinity chromatography step. RNAP-containing fractions were combined, dialyzed against 10 mM Tris/HCl (pH 7.5), 100 mM NaCl, 50% (v/v) glycerol, 0.1 mM EDTA, 0.1 mM DTT, and stored at −20 °C.

The production of RNAP for NMR studies was based on ref. [69]. Expression was carried out in E. coli BL21 (λ DE3) containing plasmid pVS10. Cells were grown in LB medium supplemented with ampicillin to an $OD_{600}$ of 0.7. The temperature was lowered to 16 °C and overexpression was induced at $OD_{600}$ = 0.8 with 0.5 mM IPTG. After overnight growth cells were harvested by centrifugation. The pellet was resuspended in buffer A$^{RNAP2}$ (50 mM Tris/HCl (pH 6.9), 500 mM NaCl, 5% (v/v) glycerol, 1 mM β-mercaptoethanol (β-ME)) containing 10 mM imidazole, DNase I (AppliChem GmbH, Darmstadt, Germany), and 1/2 protease inhibitor tablet (cOmplete, EDTA-free, Roche Diagnostics, Mannheim, Germany). Cells were lysed using a microfluidizer and the lysate was cleared by centrifugation. The supernatant was applied to a 40 ml Ni$^{2+}$-Chelating Sepharose column (GE Healthcare, Munich, Germany). After washing with buffer A$^{RNAP2}$ containing 10 mM imidazole RNAP was eluted using an imidazole gradient from 90 mM to 1 M imidazole in buffer A$^{RNAP2}$. RNAP-containing fractions were dialyzed against buffer B$^{RNAP2}$ (50 mM Tris/HCl (pH 6.9), 5% (v/v) glycerol, 0.5 mM EDTA, 1 mM β-ME) containing 100 mM NaCl and then applied to two coupled 5 ml Heparin FF columns (GE Healthcare, Munich, Germany). The columns were washed with buffer B$^{RNAP2}$ (containing 100 mM NaCl) and RNAP was eluted with a constant gradient from 100 mM to 1 M NaCl in buffer B$^{RNAP2}$. The fractions containing core RNAP were dialyzed against buffer C$^{RNAP2}$ (50 mM Tris/HCl (pH 6.9), 150 mM NaCl, 5% (v/v) glycerol, 0.5 mM EDTA, 1 mM β-ME) and subsequently concentrated by ultrafiltration. The concentrate was applied to a HiLoad S200 size exclusion column (GE Healthare, Munich, Germany) equilibrated with buffer C$^{RNAP2}$ to remove inactive RNAP aggregates. Fractions containing pure, active enzyme were concentrated by ultrafiltration, glycerol was added to a final concentration of 50% (v/v) and the protein solution was stored at −20 °C.

Protein purity was checked by SDS-PAGE, the absence of nucleic acids was checked by recording UV/Vis spectra on a Nanodrop ND-1000 spectrometer (PEQLAB, Erlangen, Germany). Concentrations were determined by measuring the absorbance at 280 nm ($A_{280}$) in a 10 mm quartz cuvette (Hellma, Müllheim, Germany) on a Biospectrometer basic (Eppendorf, Hamburg, Germany).

**Isotopic labeling.** $^{15}$N- and $^{15}$N/$^{13}$C-labeled proteins were produced by growing E. coli cells in M9 medium[70,71] containing ($^{15}$NH$_4$)$_2$SO$_4$ and $^{13}$C-D-glucose. For the production of perdeuterated proteins, cells were grown in M9 medium[70,71] prepared with increasing amounts of D$_2$O (25% (v/v), 50% (v/v), 99.9% (v/v) D$_2$O; Eurisotop, Saint-Aubin, France) with d$_7$-glucose as carbon source. The site-specific [$^1$H,$^{13}$C]-labeling of Ile, Leu, and Val methyl groups in perdeuterated proteins was performed according to published protocols[72], i.e., expression was carried out as described for the production of perdeuterated proteins, but the medium contained d$_7$-glucose as carbon source and 60 mg/l 2-keto-3-d-3-4-$^{13}$C-butyrate and 100 mg/l 2-keto-3-methyl-d$_3$-3-d$_1$-4-$^{13}$C-butyrate (both from Eurisotop, St. Aubin Cedex, France) were added 1 h prior to induction. Expression and purification were as described for the production of unlabeled proteins.

**NMR spectroscopy.** NMR experiments were performed on Bruker Avance 700 MHz, Bruker Ascend Aeon 900 MHz, and Bruker Ascend Aeon 1000 MHz spectrometers. All spectrometers were equipped with cryogenically cooled, inverse triple resonance probes. Processing of NMR data was carried out using in-house routines. 2D/3D spectra were visualized and analyzed by NMRViewJ (One Moon Scientific, Inc., Westfield, NJ, USA), 1D spectra by MatLab (The MathWorks, Inc., Version 7.1.0.183). Measurements were conducted at 15 °C. The initial sample volume was 500 µl, if not stated otherwise.

The resonance assignments for the backbone amide groups of RfaH and for the methyl groups of RfaH-CTD were taken from a previous study[31]. For resonance assignment of the RfaH methyl groups [$^{13}$C, $^{15}$N]-RfaH in 25 mM HEPES (pH 7.5), 50 mM NaCl, 5% (v/v) glycerol, 1 mM DTT, 10% D$_2$O and [I,L,V]-RfaH in 50 mM Na$_2$HPO$_4$/NaH$_2$PO$_4$ (pH 7.5), 50 mM KCl, 0.3 mM EDTA, 99.9% D$_2$O were used. The assignment was based on standard double and triple resonance experiments on [$^{13}$C,$^{15}$N]-RfaH with (H)CCH-total correlation spectroscopy (TOCSY) and H(C)CH-TOCSY spectra allowing the non-sequence-specific

identification of peaks belonging to the two methyl groups within individual Leu or Val side chains. Additionally, 3D CCH- and HCH-nuclear Overhauser effect spectroscopy (NOESY) spectra (mixing times: 250 and 200 ms, respectively) were obtained from [I,L,V]-RfaH. Combining the NOESY patterns with structural information from the crystal structure of the RfaH:ops9 complex (protein data bank (PDB) ID: '5OND'), and the identification of associated methyl groups finally allowed for the assignment of most non-overlapping resonances.

For interaction studies involving RNAP all components were in 50 mM $Na_2HPO_4/NaH_2PO_4$ (pH 7.5), 50 mM KCl, 0.3 mM EDTA, 99.9% (v/v) $D_2O$.

Interaction studies with chemical shifts changes in the fast regime on the chemical-shift timescale were analyzed by calculating the normalized chemical-shift perturbation ($\Delta\delta_{norm}$) according to Eq. (1) for [$^1H$,$^{13}C$] correlation spectra.

$$\Delta\delta_{norm} = \sqrt{\left(\Delta\delta^1H\right)^2 + \left[0.25\left(\Delta\delta^{13}C\right)\right]^2} \quad (1)$$

where $\Delta\delta$ is the resonance frequency difference in ppm.

To analyze the signal intensity quantitatively in both 1D and 2D experiments, the intensity was normalized by the concentration of the labeled protein, the number of scans, the receiver gain, and the length of the 90° proton pulse. The ratio of remaining signal intensities and signal intensities in the spectrum of the free, labeled protein was calculated for each titration step, resulting in relative signal intensities. The determination of the interaction surface of opsEC and S10 on RfaH was carried out as described in ref. [39]. In brief, the mean value of all relative signal intensities in each titration step was determined and experiment-specific thresholds of the mean value were defined. Residues with relative signal intensities below these thresholds were classified as either strongly or moderately affected and Leu and Val residues were considered as affected if at least one of the two signals showed a significant decrease in intensity. Only unambiguously assigned signals were used in the analysis. Affected residues were mapped on the three-dimensional structure of RfaH/RfaH-CTD and binding surfaces were graphically extended by (i) highlighting the complete amino acids instead of only the methyl group and (ii) highlighting the two amino acids on either side of an affected Ile, Leu, or Val residue unless they were unaffected/unassigned Ile, Leu, Val residues.

Translational diffusion coefficients (D) were determined using a stimulated echo (STE) experiment combined with a 1D [$^1H$, $^{13}C$]-HMQC for selecting $^{13}C$-bound protons using an [I,L,V]-RfaH sample in $D_2O$ buffer[73]. Gradient pulses ($\delta_{grd}$/2) for de- and rephasing were 2.5 ms and the diffusion time ($\Delta_{diff}$) was set to 80 ms. Gradient strengths (g) were varied between 1 and 47 G cm$^{-1}$. The decay of signal intensity (I) was fitted to Equation (2) using GraFit (Erithacus Software Ltd., Horley, UK, Version 6.0.12).

$$\frac{I}{I_0} = e^{-D \cdot \gamma_H^2 \cdot g^2 \cdot \delta_{grd}^2 \cdot \left(\Delta_{diff} - \frac{\delta_{grd}}{3} - \frac{\tau}{2}\right)} \quad (2)$$

with $I_0$ being the initial signal intensity, $\gamma_H$ the gyromagnetic ratio of protons, and $\tau$ the recovery delay after the gradient pulses (200 μs).

CEST experiments were carried out at 298 K and 900 MHz $^1H$-frequency according to ref. [36], using a [$^2H$, $^{13}C$, $^{15}N$]-RfaH sample in 25 mM HEPES (pH 7.5), 50 mM NaCl, 5% (v/v) glycerol, 1 mM DTT, 10% $D_2O$. Saturation was achieved by a 35 Hz $B_1$-field applied during an exchange period of 500 ms.

CPMG relaxation dispersion experiments were conducted at 288 K and a 700 MHz $^1H$-frequency using a [$^2H$, $^{15}N$]-RfaH sample in 10 mM $K_2HPO_4/KH_2PO_4$ (pH 7.5), 50 mM KCl, 10% $D_2O$. The constant time approach[74] was applied with a total constant time period of 36 ms and $\nu_{CPMG}$ ranging from 30 to 2000 Hz.

**Assembly of opsEC.** Assembly of the ops-paused EC and design of the nucleic acids were based on published methods[38]. First a RNA:DNA-hybrid was formed from the ops-template (T) DNA (Supplementary Table 2) and the ops-RNA (Supplementary Table 2). Stock solutions of both oligos (1 mM in 99.9% $D_2O$) were diluted with buffer (50 mM $Na_2HPO_4/NaH_2PO_4$ (pH 7.5), 50 mM KCl, 0.3 mM EDTA in 99.9% $D_2O$) by 1:1 and mixed at an equimolar ratio. The mixture was incubated for 1 min at 95 °C, then for 10 min at 70 °C, and finally cooled to room temperature within 15 min. RNAP (typically at 50–100 μM) was added at 1.3 molar excess over the hybrid, followed by 10 min incubation at room temperature. Finally, the NT-opsDNA strand (Supplementary Table 2; 1 mM stock solution in $D_2O$) was added at a molar ratio of 1:1.3:3 (T-ops-DNA/ops-RNA-hybrid:RNAP: NT-ops-DNA) and incubated for 10 min at 37 °C. To increase the long-term stability of the complex, 2 mM DTT, 5 mM $MgCl_2$ and 5% (v/v) $d_8$-glycerol were added to the sample.

**In vitro transcription assay.** Linear templates for in vitro transcription were made by PCR and purified via a QIAquick PCR purification kit (Qiagen, Valencia, CA). For the first-round reaction, a linear template was generated by PCR of pIA349 (Supplementary Table 3) using a top biotinylated primer and a bottom primer with an EcoRI recognition site, as described in ref. [13]. When indicated, the template was pre-incubated with a cleavage-deficient EcoRI Q111 mutant (at 3 μM; to ensure complete occupancy of the roadblock) in TGA2 (20 mM Tris-acetate, 20 mM Na-acetate, 2 mM Mg-acetate, 5% glycerol, 1 mM DTT, 0.1 mM EDTA, pH 7.9) for 15 min at 37 °C. The biotinylated DNA template (200 nM), RNAP holoenzyme (350 nM), ApU (100 μM) and 5 μM each CTP, GTP, and ATP were incubated with

prewashed Streptavidin coated magnetic beads (Dynabeads® MyOneTM Streptavidin C1) in 40 μl volume for 15 min at 37 °C to form halted G37 ECs. WT RfaH was added at 100 nM (to ensure that all RfaH was bound to the EC), followed by a 2-min incubation at 37 °C. Unlabeled NTPs (20 μM GTP, 200 μM ATP, CTP, and UTP) and rifapentin (25 μg/ml) were added for 10 min at 37 °C. The supernatant was collected using a Magnetic Separation Stand (Promega) and purified through AutoSeq G-50 spin columns (GE Healthcare) pre-equilibrated with TGA2; 25 μl/column.

For the second-round reaction, WT ops (pIA1087) or G8C ops (pZL23) templates were prepared as described in ref. [29]. The resulting linear templates contained a T7A1 promoter followed by an initial 24 nt T-less transcribed region; the run-off transcript generated on these templates is 79-nt long. Linear DNA template (30 nM), holo RNAP (50 nM), ApU (100 μM), and starting NTP subsets (1 μM CTP, 5 μM ATP and GTP, 10 μCi [α$^{32}P$]-GTP, 3000 Ci/mmol) were mixed in 100 μl of TGA2. Reactions were incubated for 15 min at 37 °C; thus halted ECs were stored on ice.

An equal volume of RfaH in TGA2 or supernatant from the first round (to yield 50 nM final concentrations) was added to the EC, followed by a 2-min incubation at 37 °C. Transcription was restarted by addition of nucleotides (5 μM GTP, 150 μM ATP, CTP, and UTP) and rifapentin to 25 μg/ml. Samples were removed at time points indicated in the figure and quenched by addition of an equal volume of STOP buffer (10 M urea, 60 mM EDTA, 45 mM Tris-borate). Samples were heated for 2 min at 95 °C and separated by electrophoresis in denaturing 9% acrylamide (19:1) gels (7 M Urea, 0.5X TBE). The gels were dried and RNA products were visualized and quantified using FLA9000 Phosphorimaging System, ImageQuant Software, and Microsoft Excel. In vitro transcription assays were carried out in triplicates and averaged.

**Model of RfaH$^{\beta CTD}$.** RfaH-NTD (PDB ID: '5OND') and RfaH-CTD in the all-β state (PDB ID: '2LCL') were superimposed on the structure of T. maritima NusG (PDB ID: '2LQ8'). No structural rearrangements were applied.

**Programs.** All molecular structures were visualized using The PyMOL Molecular Graphics System (Version 1.7, Schrödinger, LLC). Superpositions of protein and nucleic-acid structures were prepared with COOT[75]. Interaction surfaces were analyzed by the 'protein interfaces, surfaces and assemblies' service PISA at the European Bioinformatics Institute (http://www.ebi.ac.uk/pdbe/prot_int/pistart.html)[76].

## Data availability
The source data underlying Fig. 2b, Fig. 4b, Fig. 5b, Fig. 6d, Supplementary Figure 2, and Supplementary Figure 5e are provided as a Source Data file. Other data are available form the corresponding authors upon reasonable request.

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

## Acknowledgements

We thank Ramona Heißmann, and Andrea Hager for excellent technical assistance. P.K.Z. was supported by the The Elite Network Bavaria in the framework of the Elite Study Program "Macromolecular Science". The work was supported by grant Ro 617/21-1 (P.R.) from the Deutsche Forschungsgemeinschaft, the Ludwig-Schaefer award of Columbia University (P.R.), and grant GM67153 (I.A.) from the National Institutes of Health.

## Author contributions

P.K.Z., K.S., and S.H.K. carried out the NMR experiments. I.A. carried out the in vitro experiments. S.H.K., P.R., and I.A. designed and supervised research and prepared the manuscript with input from all authors.

## Additional information

**Competing interests:** The authors declare no competing interests.

