## [Peer Review File · Nature Communications]

Reviewers' comments:

Reviewer #1 (Remarks to the Author):

Manuscript by Zuber et al reports the reversible re-folding of the bacterial transcription anti-terminator RfaH, as the underlying mechanism of its toggling between the auto-inhibited and active forms. The all-alpha-to-all-beta refolding ("transformation") of RfaH CTD is not only the most profound structural transition within a single protein domain, but also the one leading to a dramatic change in its functionality. Given the frequency with which "allosteric changes" and "activation of cryptic function" are invoked in explanations of proteins alternate activity/specificity, the study of RfaH "transformation" and its implications has broad ontological impact beyond the field of transcription regulation.

Zuker et al used a combination of structural (NMR) and functional (in vitro transcription) methods to demonstrate that RfaH functional cycle (misnamed by the authors as its "life cycle") is borne out by its structural transitions. The existence of the two RfaH states and of the two distinct corresponding functionalities has been demonstrated in authors' previous work. The report under review employed time- and residue-resolved NMR-based techniques to monitor the transition between these two states as a part of the transcription cycle. Of particular importance is the demonstration that at the end of the cycle the active form of RfaH spontaneously refolds back into the autoinhibited state, providing an explanation of the earlier observation that RfaH activity in the cell is limited to the ops-containing operons after the initial recruitment and activation (the anti-termination would have otherwise spread to the rest of the genome).

NMR experiments, complemented by an array of in vitro transcription assays, fully support the conclusions put forth in the manuscript. The burden of proof is adequately met, and the experiments reported in the main and supplementary parts of the manuscript comprehensively address the potential caveats and concerns. The report is of high technical quality and relies in the large part on the state-of-the-art NMR approach allowing for the read-out of the time-resolved protein conformational changes in solution and in presence of other macromolecules. It bears noting that the authors did not explicitly solve the RfaH structures along its "transformation" pathway, but instead benchmarked the readout using well-characterized mutant forms of RfaH representing its stable structural variants (or those of its domains). This caveat does not take away from the validity of the conclusions, but in the interest of clarity has to be stated more explicitly in the text.

I recommend this manuscript for publication without major revisions, or additional experimental data, provided that specific concerns listed below are adequately addressed.

Specific comments:

1. The manuscript should be edited for semantics (e.g. the "life cycle" is not applicable to proteins in the classic meaning of the term (the reproductive cycle of an autopoietic entity), nor to the modified term, describing the cycle between protein synthesis from amino acids and its degradation into the mix of the same).
2. The design and the succession of experiments are well thought out, but the written narrative is often confounding. For example, the final 3 paragraphs (lines 223-249) is dedicated to "probing" of the RfaH "state" after it has been released from the elongation complex. Neither the "state", nor the approach to "probing" are defined. The report of an extensive in vitro transcription experimentation follows, and culminates in the conclusion which structurally has no premiss. This narrative structure would be much improved by the addition of a couple of sentences in the beginning, contrasting the two possible states and enumerating the experimental outcomes for each of them.

3. The narrative structure of the NMR part of the Results is even more rudimentary. For example, the rather extended 2nd paragraph of this section of the manuscript (lines 110-132) begins with a non sequitur about the “saturation of ^{15}N spins by a weak radio frequency field”, followed by the detailed and monotonous enumeration of experimental data, punctuated by the opaque asides to RfaH structure. In order to make this discourse assessable by the broad readership, this and the rest of the NMR experiments have to be recast at least as a minimalist logical argument (premiss, inferences/entailments, statement of data as it relates to entailments, conclusion).

4. References are generally adequate or acceptable, with the exception of reference 3. This particular reference is neither original, nor comprehensive report of the NusG role in anti-backtracking. A more prudent reference to the same effect is the earlier report - PMID: 15680325.

Reviewer #2 (Remarks to the Author):

In their paper, the authors describe a structural rearrangement in the RfaH protein. Based on state-of-the-art NMR experiments, they show how the CTD can refold from an alpha helical in to a beta sheet structure. The main insights from the paper are i) the signals/ interactions that trigger the structural transition and ii) the functional consequence of the two protein forms.

The study is well performed and provides clear insights into the biological role and mechanism of RfaH. There are some points that I would like to be addressed, but in general I think that this work is of very high interest for Nature Communications and I support publication when the points below are addressed.

What is the 45deg rotation arrow in Fig 2C? Both panels appear to have the same orientation.

Supplementary Fig 1: Why did the authors not perform the CEST and CPMG experiments on the more sensitive methyl groups?

Page 7 “Signals corresponding to the all- β RfaH-CTD could not be observed during the titration, suggesting that binding to DNA alone cannot be a signal for domain opening.” Did the authors run CEST and CPMG on the complex to see a potential transient opening? The same could hold for the interaction of RfaH with RNAP. Performing CEST and CPMG experiments on this latter complex might, however, not be feasible due to sample stability and the very large size. Nevertheless, the authors should discuss the possibility of the transient opening of RfaH upon interaction with only DNA or only RNAP. Such a transient opening could provide a role for the encounter complex (Fig. 7).

I don't understand how figure 4b is made. Is the decrease in signal intensity based on the starting spectrum of the all alpha state? Resonances of the all beta state appear during the titration, but this is probably not considered in the graph. Many of the resonances for which the intensity only decrease to a small degree might be due to the fact that for those resonances the alpha and beta states have a similar chemical shift. In that case the plot is somewhat confusing. In the spectrum residues that appear and disappear have been indicated, but there are only few situations where both the appearing and disappearing residues have been labeled. Maybe the authors can indicate both states for all residues to make clear that the disappearing and appearing resonances correlate.

How does removal of the “E48:R138” salt-bridge in RfaH influence the function of the protein, e.g. as assayed in Figure 6?

There are some formatting errors in my PDF viewer. E.g. bottom of page 22, the gyromagnetic ratio symbol (line 477) are indicated as boxes.

Dear Reviewers,

we thank you for your efforts and we appreciate your insightful comments, which will enhance the manuscript considerably. Below, we provide detailed responses to your comments and explain how we addressed them in a revised version of our manuscript. All changes in the manuscript file are highlighted in red.

Reviewer #1

1. The manuscript should be edited for semantics (e.g. the “life cycle” is not applicable to proteins in the classic meaning of the term (the reproductive cycle of an autopoietic entity), nor to the modified term, describing the cycle between protein synthesis from amino acids and its degradation into the mix of the same).

We agree with the reviewer and replaced the term „life cycle“ by „functional cycle“.

2. The design and the succession of experiments are well thought out, but the written narrative is often confounding. For example, the final 3 paragraphs (lines 223-249) is dedicated to “probing” of the RfaH “state” after it has been released from the elongation complex. Neither the “state”, nor the approach to “probing” are defined. The report of an extensive in vitro transcription experimentation follows, and culminates in the conclusion which structurally has no premiss. This narrative structure would be much improved by the addition of a couple of sentences in the beginning, contrasting the two possible states and enumerating the experimental outcomes for each of them.

We added leading sentences stating the rationale for the presented in vitro experiments and the expected outcomes. We make it clear that while the NMR experiments clearly show that RfaH can transform back from the open state with the all- β CTD into the autoinhibited form with the all- α CTD, the in vitro assay provides alternative (and complementary) means to assess whether RfaH returns into its autoinhibited state under more physiological conditions, but does not provide direct structural information.

3. The narrative structure of the NMR part of the Results is even more rudimentary. For example, the rather extended 2nd paragraph of this section of the manuscript (lines 110-132) begins with a non sequitur about the “saturation of ^{15}N spins by a weak radio

frequency field”, followed by the detailed and monotonous enumeration of experimental data, punctuated by the opaque asides to RfaH structure. In order to make this discourse assessable by the broad readership, this and the rest of the NMR experiments have to be recast at least as a minimalist logical argument (premiss, inferences/entailments, statement of data as it relates to entailments, conclusion).

We carefully revised the narrative structure of the Results. Each section starts with the question to be targeted/answered, followed by the description of the approach/setup, the discussion of possible outcomes (if necessary), the interpretation of the results, and finally the conclusion. In particular, the description of the CEST and CPMG experiments (lines 104 ff) has been extended to be easily understandable by the broad readership.

4. References are generally adequate or acceptable, with the exception of reference 3. This particular reference is neither original, nor comprehensive report of the NusG role in anti-backtracking. A more prudent reference to the same effect is the earlier report - PMID: 15680325.

We agree with the reviewer and replaced reference 3 by „Bar-Nahum, G. et al. A ratchet mechanism of transcription elongation and its control. Cell 120, 183–193 (2005)“.

Reviewer #2

1. What is the 45deg rotation arrow in Fig 2C? Both panels appear to have the same orientation.

The arrow indicates that the surface representation is rotated by 45 ° ccw with respect to the ribbon representation. We added this information to the figure legend.

2. Supplementary Fig 1: Why did the authors not perform the CEST and CPMG experiments on the more sensitive methyl groups?

We agree with the reviewer that CEST and CPMG experiments of methyl groups would be more sensitive. However, the data quality of experiments with ¹⁵N-labeled samples was sufficient to exclude the presence of the open state with the CTD in the β-barrel conformation in free RfaH (within the general limits of the experiments). In particular, in none of the CEST profiles of free RfaH a second dip at the resonances corresponding to the β-barrel state could be observed.

3. Page 7 “Signals corresponding to the all-β RfaH-CTD could not be observed during the titration, suggesting that binding to DNA alone cannot be a signal for domain opening.” Did the authors run CEST and CPMG on the complex to see a potential transient opening? The same could hold for the interaction of RfaH with RNAP. Performing CEST and CPMG experiments on this latter complex might, however, not be feasible due to sample stability and the very large size. Nevertheless, the authors should discuss the possibility of the transient opening of RfaH upon interaction with only DNA or only RNAP. Such a transient opening could provide a role for the encounter complex (Fig. 7).

We carried out CEST experiments for ²H,¹⁵N-RfaH in the presence of opsDNA, but unfortunately the relaxation properties of the complex were not sufficient to obtain reasonable spectra (broad signals). Analogous experiments for the RfaH:RNAP complex might not be possible due to insufficient sensitivity, even if methyl groups were used as NMR probes, and probably sample instability over the long measuring time (estimated experimental time would be on the order of several weeks).

Our present findings indicate that only the ops-paused EC is able to relieve autoinhibition and that neither RNAP nor DNA alone is able to activate RfaH. In a physiological context single-stranded opsDNA does not occur outside the EC and in the ops-paused EC the ops hairpin is exposed on the surface of RNAP so that RfaH has to interact with certain RNAP elements as soon as it establishes contacts with the ops NT-DNA strand. Thus, binding to DNA alone or RNAP alone do not represent physiological situations so that we did not

establish methyl group-based CEST/CPMG experiments for these binary systems. However, we argue that recruitment might involve an encounter complex where RfaH establishes contacts to both opsDNA and certain RNAP elements (most probably the β GL) and that these interactions may influence the RfaH interdomain interactions. We extended the role of the encounter complex in the discussion and discuss the possibility that DNA or RNAP alone could weaken the NTD:CTD interface.

CEST and CPMG experiments on the encounter complex might give valuable insights into changes in the stability of the autoinhibited state of RfaH, but here we will meet the same problems as for the RfaH:RNAP complex. Thus, we will try to identify and characterize the encounter complex by other means in the future.

4. I don't understand how figure 4b is made. Is the decrease in signal intensity based on the starting spectrum of the all alpha state?

In a certain titration step relative intensities were determined as ratio of remaining signal intensities and signal intensities in the spectrum of free [I,L,V]-RfaH. We added a brief description of the quantitative analysis to the main text and a detailed description in the Methods section.

Resonances of the all beta state appear during the titration, but this is probably not considered in the graph. Many of the resonances for which the intensity only decrease to a small degree might be due to the fact that for those resonances the alpha and beta states have a similar chemical shift. In that case the plot is somewhat confusing.

Resonances corresponding to the all- β CTD were indeed not considered in the quantitative analysis. To prevent confusion, we now limited the analysis to the RfaH-NTD, in particular as the decrease of RfaH-CTD signals in the all- α state is due to domain separation and refolding and not relevant for the analysis of the RfaH-NTD:opsEC binding surface. Figures 4b and c were adapted accordingly.

In the spectrum residues that appear and disappear have been indicated, but there are only few situations where both the appearing and disappearing residues have been labeled. Maybe the authors can indicate both states for all residues to make clear that the disappearing and appearing resonances correlate.

In Figure 4a we want to showcase signals corresponding to RfaH-NTD, RfaH-CTD in the all- α state and RfaH-CTD in the all- β state, but we agree with the reviewer that it would be valuable to have more CTD signals in both states labeled. To avoid confusing labeling

in Figure 4a we added a new supplementary figure (Fig. S5) where we show methyl-TROSY spectra of free [I,L,V]-RfaH and [I,L,V]-RfaH:opsEC that contain all assigned CTD signals in the all- α and in the all- β state.

5. How does removal of the “E48:R138” salt-bridge in RfaH influence the function of the protein, e.g. as assayed in Figure 6?

Weakening or elimination of the salt bridge reduces/abolishes RfaH dependence on the ops element, i.e. it turns RfaH into a NusG-like general transcription factor. This information has been added to the text. As this variant was not tested in the in vitro assay on the G8C template, we believe that a discussion of this variant in this context would not give any added value at this point. Instead, we cite published data showing that RfaH substitutions at the interdomain interface designed to convert it into NusG (I93E and F130V) promote domain dissociation and allow RfaH recruitment at the G8C template.

6. There are some formatting errors in my PDF viewer. E.g. bottom of page 22, the gyromagnetic ratio symbol (line 477) are indicated as boxes.

We carefully checked the manuscript and corrected any formatting errors.

REVIEWERS' COMMENTS:

Reviewer #2 (Remarks to the Author):

The authors have fully addressed my comments and the comments of the other reviewer.

I congratulate them on this very nice paper.